# Characterization of Sarcopenia in an IBD Population Attending an Italian Gastroenterology Tertiary Center

**DOI:** 10.3390/nu11102281

**Published:** 2019-09-24

**Authors:** Marco Pizzoferrato, Roberto de Sire, Fabio Ingravalle, Maria Chiara Mentella, Valentina Petito, Anna Maria Martone, Francesco Landi, Giacinto Abele Donato Miggiano, Maria Cristina Mele, Loris Riccardo Lopetuso, Elisa Schiavoni, Daniele Napolitano, Laura Turchini, Andrea Poscia, Nicola Nicolotti, Alfredo Papa, Alessandro Armuzzi, Franco Scaldaferri, Antonio Gasbarrini

**Affiliations:** 1UOC Medicina Interna e Gastroenterologia, Fondazione Policlinico Universitario Agostino Gemelli IRCCS, 00168 Roma, Italy; valepetito88@gmail.com (V.P.); lopetusoloris@libero.it (L.R.L.); elisa.schiavoni90@gmail.com (E.S.); daniele.napolitano@policlinicogemelli.it (D.N.); turchini.laura@gmail.com (L.T.); andrea.poscia@unicatt.it (A.P.); alfredo.papa@unicatt.it (A.P.); alessandro.armuzzi@unicatt.it (A.A.); antonio.gasbarrini@unicatt.it (A.G.); 2Istituto di Patologia Speciale Medica, Università Cattolica del Sacro Cuore, 00168 Roma, Italy; fabio.ingravalle@gmail.com; 3UOC Nutrizione Clinica, Fondazione Policlinico Universitario Agostino Gemelli IRCCS, 00168 Roma, Italy; mariachiara.mentella@libero.it (M.C.M.); giacintoabele.miggiano@unicatt.it (G.A.D.M.); mariacristina.mele@unicatt.it (M.C.M.); 4UOC Geriatria, Fondazione Policlinico Universitario Agostino Gemelli IRCCS, 00168 Roma, Italy; annamaria.martone@guest.policlinicogemelli.it (A.M.M.); francescolandi@policlinicogemelli.it (F.L.); 5Medical Management, Fondazione Policlinico Universitario A. Gemelli IRCCS, 00168 Roma, Italy; nicola.nicolotti@policlinicogemelli.it

**Keywords:** IBD, sarcopenia, inflammation, nutrition

## Abstract

(1) Background: There is growing interest in the assessment of muscular mass in inflammatory bowel disease (IBD) as sarcopenia is associated with important outcomes. The aim of the study was to evaluate the percentage of sarcopenia in IBD patients, characterizing methods for assessment and clinical symptoms associated to it. (2) Methods: Consecutive IBD patients accessing the Fondazione Policlinico Agostino Gemelli Istituto di Ricovero e Cura a Carattere Scientifico (IRCCS) were enrolled. Healthy patients, elderly or elderly sarcopenic patients, were enrolled as controls. Skeletal muscle mass was evaluated by Dual Energy X-ray Absorptiometry (DEXA) or Bio-Impedensometric Analysis (BIA). Asthenia degree was assessed by subjective visual analogue scales (VAS). Quality of life was measured by the EQ-5D questionnaire. (3) Results: Patients with IBD showed a significant reduction in skeletal muscle mass than healthy controls with lower DEXA and BIA parameters. Moreover, IBD patients presented a lower perception of muscle strength with a higher incidence of asthenia and reduction in quality of life when compared with healthy controls. A significant association between loss in skeletal muscle mass and high asthenia degree was found, configuring a condition of sarcopenia in about one third of patients with IBD. (4) Conclusions: Sarcopenia is common in IBD patients and it is associated with fatigue perception as well as a reduction in quality of life. Therefore, routine assessment of nutritional status and body composition should be a cornerstone in clinical practice, bringing gastroenterologists and nutritionists closer together for a compact, defined picture.

## 1. Introduction

Sarcopenia is defined as a progressive and generalized skeletal muscle disorder characterized by a significant reduction in skeletal muscle mass associated with low muscle strength and low physical performance. Sarcopenia can be primary (age-related) or secondary to inactivity, systemic diseases (inflammatory, neoplastic or organ failure) or malnutrition [1]. The diagnosis of sarcopenia requires the demonstration of a significant reduction in muscle mass associated with a reduction of muscle function. Muscular mass can be studied with radiologic methods, in particular magnetic resonance imaging (MRI) and computed tomography (CT), which are considered the gold standard for non-invasive assessment. However, higher costs, lack of portability, necessity of trained personnel and exposition to X-rays and contrast medium strongly limit the use of MRI and CT in clinical practice. On the contrary, Dual Energy X-ray Absorptiometry (DEXA) and Bio-Impedensometric analysis (BIA) are considered punctual, affordable, widely available and portable techniques to assess skeletal muscle mass. Muscular function can be studied with a specific test evaluating hand grip or chain stand [2,3].

Inflammatory bowel diseases (IBD) are often associated with malnutrition and significant alteration of body composition [4,5]. Major determinants of malnutrition in IBD patients are reduction of caloric intake, malabsorption, side effects of drugs and increases in basal energy expenditure due to inflammation with significant alteration in lipid and carbohydrate metabolism [6]. Malnutrition in IBD is characterized by weight loss during the acute phase of disease followed by a gradual recovery during the disease remission [7]. It was also shown that patients with IBD demonstrated a reduction of skeletal muscle and bone masses with expansion of visceral and “creeping” fat [8]. Moreover, recent evidence demonstrated a significant reduction in skeletal muscle in IBD patients when compared with healthy controls, with an incidence between 36.7% and 65% of patients [9]. Generally, sarcopenic patients presented a lower median body mass index (BMI) than patients with a normal body composition [10], but it was also shown that 40% of sarcopenic patients have a normal BMI and up to 20% are overweight or obese [11]. Unfortunately, none of the available studies assessed both the anatomical and functional components required for the correct diagnosis of sarcopenia. 

Concerning the reduction on skeletal muscle mass (sarcopenia) in IBD patients, the exact pathophysiological mechanisms have not yet been completely defined. Recent studies have demonstrated the crucial role of the Insulin-like Growth Factor 1/ PhosphoInositide 3 Kinase/ Protein Kinase B/ Mammalian Target of Rapamycin (IGF1/PI3K/Akt/mTOR) axis dysregulation associated with reduction of IGF1-R in muscle specimens of patients with IBD [12]. 

Therefore, IBD can be considered as a cause of secondary sarcopenia but, unfortunately, strong evidence is not yet available due to the lack of homogeneous studies with appropriate sample numbers. Moreover, standardized and validated diagnostic criteria for sarcopenia in patients with IBD are currently not available [13]. 

The aim of the study was to evaluate the percentage of sarcopenia in IBD patients, characterizing methods for assessment and clinical symptoms associated to it.

## 2. Materials and Methods 

### 2.1. Study Design

Four cohorts were enrolled in the study: A first cohort was composed of patients with confirmed diagnosis of Crohn’s disease (CD) and ulcerative colitis (UC), attending the Digestive Disease Center (CEMAD) at Fondazione Policlinico Agostino Gemelli Istituto di Ricovero e Cura a Carattere Scientifico (IRCCS); healthy controls were enrolled in the second cohort; a third cohort was composed of healthy elderly; and a fourth cohort was composed of elderly with primary sarcopenia, attending the Center of Aging Medicine (CEMI) at Fondazione Policlinico Agostino Gemelli IRCCS. Skeletal muscle mass was assessed with one of the following methods: Dual Energy X-ray Absorptiometry (DEXA) or Bio-Impedance Analysis (BIA). All subjects gave their informed consent for inclusion before they participated in the study. The study was conducted in accordance with the Declaration of Helsinki, and the protocol was approved by the Ethics Committee of Fondazione Policlinico Universitario Agostino Gemelli IRCCS (Prot. N.286983/19, ID: 2673 and Prot.N. P/491/CE/2011). DEXA and BIA are currently used in clinical practice for multidisciplinary evaluation. 

### 2.2. Population

A total of 127 consecutive patients from 1 February 2017 until 31 May 2017, with a confirmed diagnosis of Crohn’s disease and ulcerative colitis, attending the Digestive Disease Center (CEMAD) at Fondazione Policlinico Agostino Gemelli IRCCS were enrolled in the first cohort: 39% were women and 61% were men, with a median age of 41.60 ± 13.76 years. The median Body Mass Index (BMI) of the population was 24.63 ± 4.38 kg/m^2^. A total of 69 patients (64%) had a previous diagnosis of CD with ileal localization in 29% of patients and ileo-colic localization in 27% of patients. A total of 58 patients (46%) had a previous diagnosis of UC (Ulcerative Colitis): 25% of patients with UC presented pancolitis, 12% left colitis and 7% proctitis. Disease activity was assessed using the Harvey–Bradshaw Index (HBI) for patients with a CD and c-MAYO score per UC patients: The median HBI was 4.46 in CD patients while the median c-MAYO score was 4.8 in UC patients. A total of 70 patients (55%) were in the remission phase of disease, 30 patients (24%) had mild disease, 22 patients (17%) presented moderate disease while only 5 patients (4%) presented severe disease. A total of 58 patients (46%) were treated with corticosteroids, 67 patients (54%) with anti-TNFα and 43 patients (34%) were treated with corticosteroids + anti-TNFα. A total of 34% of the patients had previous surgery (Table 1).

A total of 89 healthy controls attending the Digestive Disease Center (CEMAD) at Fondazione Policlinico Agostino Gemelli IRCCS were enrolled in the second cohort: 49 were women (55%) and 40 were men (44%), with a median age of 37.45 ± 12.86 years and a median BMI of 25.10 ± 3.50 (kg/m^2^) (Table 1). These patients were mostly attending the endoscopy clinic.

In the third cohort, 14 healthy elderly, attending the Center of Aging Medicine (CEMI), were enrolled: 79% were women and 21% were men, with a median age of 79.86 ± 4.37 years and a median BMI of 25.74 ± 3.36 kg/m^2^. The fourth cohort was composed of 5 elderly with diagnosed primary sarcopenia, of which 80% were women and 20% were men, with a median age of 78.80 ± 4.76 years and a median BMI of 23.55 ± 3.71 kg/m^2^ (Table 1).

### 2.3. Assessment of Asthenia Degree

The degree of asthenia was measured by a specific survey administered to patients and controls. Asthenia degree was evaluated through a specific survey composed by three Visual-Analogic-Scales (VAS) aimed to measure (1) perceived actual muscular strength, (2) subjective maximal muscular strength, and (3) perceived wellness degree. Each VAS had a 0–100 range. In particular, 0 represented worst perception, 100 the best perception, and 70 was used as the threshold to define low perception. This value was chosen through ROC curves.

### 2.4. Assessment of Body Composition

Weight and height have been evaluated in all subjects enrolled with a scale equipped with a stadiometer (Seca^®^). Every subject was evaluated fasting, early in the morning and without clothes. BMI was calculated based on the measured weight and height. Muscular skeletal mass was evaluated with two methods: Dual Energy X-ray Absorptiometry (DEXA; Hologic^®^) and Bio-Impedance Analysis (BIA; Hassern^®^). Subjects were evaluated without clothes, in the supine position, with distanced and alienated limbs, making sure that they were well spaced between them. For the duration of both evaluations the patients remained motionless. Concerning DEXA, the Appendicular Skeletal Mass Index (ASMI, appendicular mass in kg/square of height in m^2^) was calculated. European Working Group on Sarcopenia in Older People (EWGSOP) criteria were used to define a significant reduction in muscular mass: ASMI < 7.23 kg/m^2^ for men and ASMI < 5.67 kg/m^2^ for women were considered effective markers of muscular mass depletion. Concerning BIA, the Skeletal Muscle Index (SMI, estimated muscular mass in kg/square of height in m^2^) was used to define skeletal muscle mass depletion. The estimation of muscular mass was obtained through the prediction equation provided by the manufacturer of the Hassern^®^ machine. EWGSOP criteria were used to define a significant reduction in muscular mass: SMI < 10.75 kg/m^2^ per for men and SMI < 6.75 kg/m^2^ for women identified muscular depletion.

### 2.5. Quality of Life Evaluation

Quality of life was measure through standard EQ-5D questionnaire (EuroQol Group^®^), which explored 5 items: Mobility, self-care, usual activities, pain/discomfort and anxiety/depression. The respondents self-rate their level of severity for each dimension using three-level (EQ-5D-3L).

### 2.6. Statistical Analysis

Results were reported in mean ± standard deviation (DS) for continuous variables with a normal distribution, while dichotomous variables were expressed in percentage form. Comparisons between the four variables were made with Pearson’s X2 exact test. The performed tests were considered significant for a value of *p* < 0.05.

## 3. Results

Increased asthenia, a reduction in skeletal muscle mass and an increase in sarcopenia were observed in IBD patients. Perceived actual muscular strength in patients with IBD was 62.45 ± 22.25, in healthy controls it was 72.27 ± 20.97, in healthy elderly it was 59.28 ± 22.94 while in sarcopenic elderly it was 51.00 ± 19.49 (*p* < 0.05). Considering the reference cut-off, 65% of patients with IBD has a reduced score compared with 33% of healthy controls, 71% of healthy elderly and 100% of sarcopenic elderly, and this finding was statistically significant (*p* < 0.05) (Figure 1b). Regarding the perceived wellness degree, IBD patients had a median value of 63.79 ± 25.25, compared with 76.68 ± 22.35 for healthy controls, 61.25 ± 25.77 for healthy elderly and 66.25 ± 18.87 for sarcopenic controls with a statistically significant difference (*p* < 0.05). 

Patients with IBD have a median BMI of 24.63 ± 4.38, while the BMI of healthy controls, elderly controls and sarcopenic elderly was, respectively, 25.10 ± 3.50, 25.74 ± 3.36 and 23.55 ± 3.71. Patients with IBD showed a significant reduction in skeletal muscle mass, in particular when analyzed with DEXA, and the ASMI value was 6.88 ± 1.13 kg/m^2^ compared with 7.02 ± 1.14 kg/m^2^ for healthy controls, 6.53 ± 0.89 kg/m^2^ for healthy elderly and 5.95 ± 0.77 kg/m^2^ for sarcopenic controls (*p* < 0.05). On the contrary, when skeletal muscular mass was assessed with BIA, SMI was 9.58 ± 7.61 kg/m^2^ in patients with IBD compares with 9.44 ± 1.83 kg/m^2^ of healthy controls, 6.99 ± 0.60 kg/m^2^ for healthy elderly and 6.27 kg/m^2^ for sarcopenic elderly (*p* < 0.05) (Table 2). Therefore, a significant depletion of skeletal muscle mass was observed in 36% of patients with IBD compared with 10% of healthy controls, 43% of healthy elderly and 100% of sarcopenic controls (*p* < 0.05) (Figure 1a). Surprisingly, only the BIA phase angle was influenced by disease activity, while there was no statistical correlation between ASMI, SMI and skeletal muscle mass and disease activity (Table 3 and Table 4). Furthermore, BMI was not different in IBD and controls.

A significant association between asthenia degree and skeletal muscle mass depletion, which can be defined as sarcopenia, was found in 28% of patients with IBD, compared with in 1% of healthy controls, in 36% of healthy elderly and in 100% of sarcopenic elderly (*p* < 0.05) (Figure 1c). Moreover, 49% of IBD patients affected with sarcopenia were treated with corticosteroids.

Quality of life alteration in IBD and sarcopenic patients. Patients with IBD showed a significant reduction three of five items explored in the EQ-5D questionnaire when compared with the healthy controls: Mobility, usual activities and pain/discomfort (*p* < 0.05); no significant difference was found for the item anxiety/depression (Figure 2).

## 4. Discussion

Sarcopenia is a progressive and generalized syndrome characterized by the loss of skeletal muscle mass and muscle strength with adverse outcomes, such as frailty, poor quality of life and mortality [13]. The pathogenesis of muscle wasting includes several elements, such as aging, systemic inflammation, mitochondrial dysfunction, increased proteolysis, decreased proteosynthesis and insulin resistance [14]. In the past, sarcopenia was considered a geriatric syndrome [15]; nowadays, instead, it is associated also to several diseases in young adults, like IBD [16]. Sarcopenia is common in IBD patients and it is possible to use it like a predictive marker for surgical intervention. Furthermore, it is associated with an increased risk of major postoperative complications [9].

This study has evaluated the incidence of sarcopenia in an IBD population attending a Gastroenterology Tertiary Center in Italy. Several studies demonstrated lower skeletal muscle mass in IBD patients than healthy controls, with a strict association with BMI and disease activity, but none of the available studies has evaluated the functional component of sarcopenia [9]. In this study the authors have evaluated the skeletal muscle mass with DEXA and bioimpedance methods while the functional aspects were evaluated with a subjective assessment of asthenia degree. In particular, perceived actual muscular strength, subjective maximal muscular strength and perceived wellness degree were studied. Hand grip and chair stand tests were not used, because it was demonstrated in our case history that a significant association between these tests and the subjective assessment of asthenia degree with VAS scales exists in IBD populations. This idea is corroborated by the observation that the association between objective radiologic bioimpedance measurements and subjective VAS scales identifies all the elderly patients in the fourth cohort with a previous diagnosis of sarcopenia (performed with objective anatomical and functional muscle evaluation).

Moreover, quality of life was evaluated in IBD and sarcopenic patients. 

The study confirmed a significant reduction in skeletal muscle mass in IBD patients compared to healthy controls, but a weak correlation with BMI, median SMI and disease activity was observed. Not significant difference was observed between IBD patients and elderly controls. BMI cannot be used in nutritional assessments of IBD patients, while DEXA and BIA should be used in clinical settings to assess body composition and measure skeletal muscle mass. DEXA and BIA can both be considered valid methods in the assessment of skeletal muscle mass, but ASMI seems to be more objective than SMI due to the lack of dependence on age (only BMI and genre) (Figure 2c). Moreover, a median SMI should be not considered due to a higher standard deviation of the data. Surprisingly, only angle phase strictly correlated with the disease activity. On one hand, phase angle seems to be a good nutritional indicator and a predictor of disease severity and therapy response. On the other hand, phase angle could be too influenced by inflammation and abdominal tissue edema, losing precision in the evaluation of skeletal muscle mass. 

Concerning the functional aspects, IBD patients presented a lower perception of muscle strength with higher incidence of asthenia when compared with healthy controls, as confirmed in a recent study [17]. Furthermore, a significant association between loss in skeletal muscle mass and high asthenia degree was found, configuring a condition of sarcopenia in about one third of patients with IBD. Currently, the use of a subjective assessment of asthenia degree is limited to the screening of sarcopenic patients to address to nutritional evaluation, but a subjective measurement of muscular strength with VAS scales could be used also to assess the function of muscle, instead of other unvalidated tests in specific populations (as in IBD patients). Therefore, the association between an objective measurement of skeletal muscle mass (with DEXA or BIA) and a subjective assessment of asthenia degree could be a valid criterion for diagnosis of sarcopenia. Furthermore, a significant association between IBD, sarcopenia and reduction of quality of life was found, especially for IBD patients with sarcopenia and relapsing disease.

The major limitations of the study are related to the significant variability of the IBD cohort, concerning, in particular, disease activity and localization, the lack of patient stratification based on drug therapy, the small number of patients who underwent both muscle assessment techniques (DEXA and BIA), and the failure to demonstrate complete correlation between the two methods (DEXA and BIA). Moreover, a functional muscle assessment was performed with subjective tests, not specifically elaborated for IBD population, which can be significantly influenced by the psychological state of patients. However, this study could represent a first step to study sarcopenia as an association between anatomical and functional dysfunction in skeletal muscle mass, in a cohort of patients with IBD in clinical practice.

## 5. Conclusions 

In conclusion, sarcopenia is a common problem in patients with IBD and a significant nutritional aspect in the clinical management of IBD. Sarcopenia correlates with an increased rate of major disease and post-operative complications; it is associated with a higher mortality and morbidity in IBD patients. Therefore, routine assessment of nutritional status and body composition should be a cornerstone in clinical practice. Indeed, universal assessment of skeletal muscle mass with BIA or DEXA and muscle strength with subjective assessment of asthenia with a VAS scale could be a valid intervention to diagnose sarcopenia, to establish perioperative interventions, reducing complications and increasing clinical outcomes. For this reason, a multidisciplinary assessment of IBD patients is necessary, bringing gastroenterologists and nutritionists closer together for a compact, defined picture.

## Figures and Tables

**Figure 1 nutrients-11-02281-f001:**
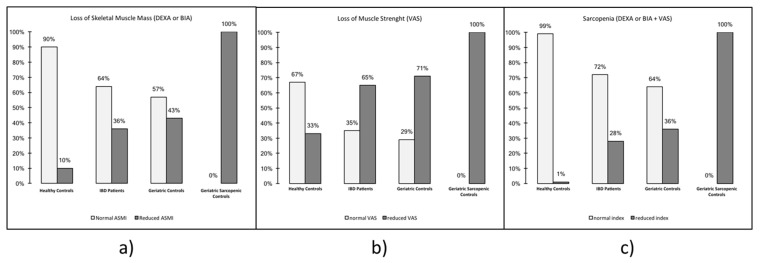
Percentage of sarcopenia in the study population: IBD patients, healthy controls, geriatric controls and geriatric sarcopenic controls. (**a**) Loss of skeletal muscle mass, (**b**) loss of muscle strength, and (**c**) loss of skeletal muscle mass and muscle strength: sarcopenia. Abbreviations. IBD: Inflammatory Bowel Disease; DEXA: Dual Energy X-ray Absorptiometry; BIA: Bio-Impedensometric Analysis. VAS: Visual Analogue Scales. ASMI: Appendicular Skeletal Mass Index.

**Figure 2 nutrients-11-02281-f002:**
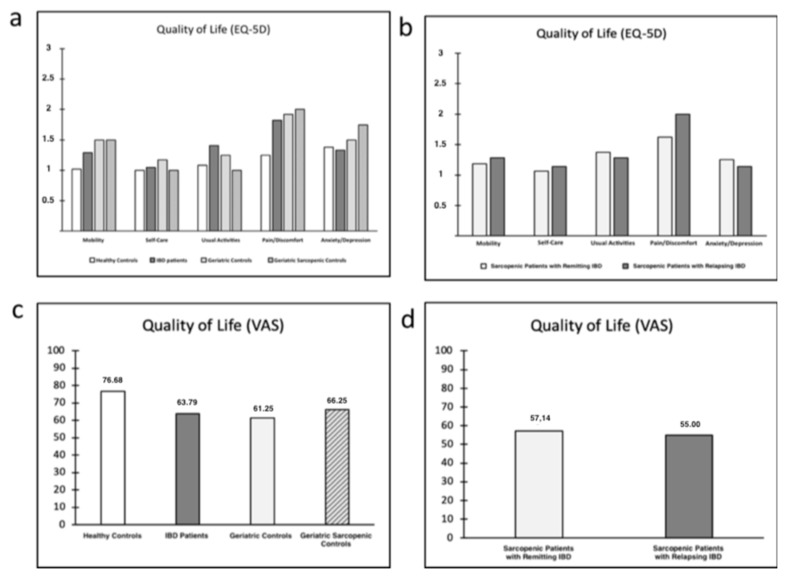
Sarcopenia and quality of life: (**a**) EQ-5D—study population; (**b**) EQ-5D—sarcopenic patients with remitting and relapsing IBD; (**c**) visual analogue scales (VAS)—study population; and (**d**) VAS—sarcopenic patients with remitting and relapsing IBD. Abbreviations. IBD: Inflammatory Bowel Disease; VAS: Visual Analogue Scale.

**Table 1 nutrients-11-02281-t001:** Patients Characteristics.

	IBD Patients	UC Patients	CD Patients	Healthy Controls	Getriac Controls	Geriatric Sarcopenic Controls
**Patient Demographics**						
**Cases, No.**	127 (100%)	58 (46%)	69 (54%)	89 (100%)	14 (100%)	5 (100%)
**Age, y**	41.60 ± 13.76	43.47 ± 12.57	40.11 ± 14.44	37.45 ± 12.86	79.86 ± 4.37	78.80 ± 4.76
**Sex, F/M**	39%/61%	17%/26%	22%/35%	55%/44%	79%/21%	80%/20%
**BMI, kg/m^2^**	24.63 ± 4.38	23.94 ± 4.74	25.2 ± 4.01	25.10 ± 3.50	25.74 ± 3.36	23.55 ± 3.71
**Extent of Disease**						
**Ileal**	29%	-	29%	-	-	-
**Ileo-Colonic**	27%	-	27%	-	-	-
**Pancolitis**	25%	25%	-	-	-	-
**Descending Colon**	12%	12%	-	-	-	-
**Rectal**	7%	7%	-	-	-	-
**Clinical Score**						
**Clinical Mayo**	-	4.8	-	-	-	-
**Harvey Bradshaw**	-	-	4.64	-	-	-
**Disease Activity**						
**Remission**	55%	22%	33%	-	-	-
**Mild**	24%	11%	13%	-	-	-
**Moderate**	17%	8%	9%	-	-	-
**Severe**	4%	2%	2%	-	-	-
**Resection, No.**						
	34%	8%	26%	-	-	-
**Therapy**						
**Steroids**	43%	20%	23%	-	-	-
**Anti-TNFα**	80%	33%	47%	-	-	-
**Skeletal Muscle Mass Evaluation**						
**DEXA**	38%	19%	19%	56%	100%	100%
**BIA**	62%	25%	37%	44%	29%	20%

Abbreviations. IBD: Inflammatory Bowel Disease; UC: Ulcerative Colitis; CD: Crohn’s Disease; BMI: Body Mass Index; DEXA: Dual Energy X-ray Absorptiometry; BIA: Bio-Impedensometric Analysis.

**Table 2 nutrients-11-02281-t002:** Biometric evaluation: parameters by measurement device.

	Measurement Device	Mean Whole Population (Std.Dev)	Mean UC Patients (Std.Dev)	Mean CD Patients (Std.Dev)	Mean Healthy Controls (Std.Dev)	Mean Geriatric Controls (Std.Dev)	Mean Geriatric Sarcopenic Controls (Std.Dev)	*p*	Male	Female	*p*
**Phase Angle**	BIA	5.568 (1.019)	5.491 (1.138)	5.618 (1.014)	5.712 (0.880)	4.340 (0.629)	4.770 (0.001)	0.1024	5.926 (0.132)	5.206 (0.094)	0.00001
**Skeletal Muscle** **Mass**	BIA	28.076 (7.613)	28.169 (7.587)	29.77 (6.463)	27.608 (7.957)	15.945 (1.388)	14.300 (0.00001)	0.0023	34.394 (0.517)	21.664 (0.486)	0.00001
**SMI**	BIA	9.587 (7.613)	9.600 (1.951)	10.005 (1.811)	9.444 (1.838)	6.992 (0.607)	6.270 (0.0001)	0.0101	11.045 (0.146)	8.108 (0.148)	0.00001
**Appendicular Lean Mass**	DEXA	19.403 (7.613)	19.936 (4.591)	19.463 (4.714)	20.402 (4.782)	15.745 (3.242)	14.888 (2.855)	0.0027	23.370 (0.399)	15.494 (0.227)	0.00001
**ASMI**	DEXA	6.886 (1.133)	6.885 (1.188)	6.622 (1.051)	7.201 (1.148)	6.529 (0.892)	5.959 (0.772)	0.0296	7.678 (0.115)	6.106 (0.080)	0.00001

Abbreviations. IBD: Inflammatory Bowel Disease; UC: Ulcerative Colitis; CD: Crohn’s Disease; DEXA: Dual Energy X-ray Absorptiometry; BIA: Bio-Impedensometric Analysis. SMI: Skeletal Muscle Index; ASMI: Appendicular Skeletal Mass Index.

**Table 3 nutrients-11-02281-t003:** Biometric evaluation: age, BMI and genre influence.

	Phase Angle	Coeff.	Std. Err.	t	*p*	95% Conf. Interval
**Age**	−0.029	0.007	−3.780	0.000	−0.045	−0.014
**BMI**	0.076	0.220	3.430	0.001	0.032	0.121
**Genre**	0.510	0.210	2.430	0.017	0.092	0.928
**Type**	0.043	0.148	0.290	0.772	−0.252	0.338
	**SMI**	**Coeff.**	**Std. Err.**	**t**	***p***	**95% Conf. Interval**
**Age**	−0.017	0.006	−2.940	0.004	−0.029	−0.006
**BMI**	0.244	0.017	14.730	0.000	0.211	0.277
**Genre**	2.449	0.155	15.760	0	2.140	2.759
**Type**	0.043	0.148	0.290	0.772	−0.252	0.338
	**ASMI**	**Coeff.**	**Std. Err.**	**t**	***p***	**95% Conf. Interval**
**Age**	−0.014	0.008	−1.700	0.095	−0.032	0.002
**BMI**	0.166	0.031	5.310	0.000	0.103	0.229
**Genre**	1.248	0.214	5.830	0	0.818	1.677
**Type**	−0.075	0.191	−0.390	0.697	−0.459	0.309

Abbreviations. BMI: Body Mass Index.

**Table 4 nutrients-11-02281-t004:** Biometric evaluation: impact of disease activity.

	Measurement Device	Mean IBD Patients in Remission (Std.Dev)	Mean IBD Patients in Mild Activity (Std.Dev)	Mean IBD Patients in Moderate Activity (Std.Dev)	Mean IBD Patients in Severe Activity (Std.Dev)	*p*
**Phase Angle**	BIA	5.919 (0.720)	5.535 (1.254)	5.244 (1.178)	4.719 (1.080)	0.0213
**Skeletal Muscle Mass**	BIA	30.706 (7.010)	29.305 (5.750)	27.216 (7.167)	25.354 (9.131)	0.1565
**SMI**	BIA	10.080 (1.620)	10.041 (1.712)	9.298 (2.165)	9.759 (2.628)	0.3993
**Appendicular Lean Mass**	DEXA	20.893 (4.716)	17.400 (3.187)	18.883 (4.174)	11.768 (0.0001)	0.0221
**ASMI**	DEXA	6.976 (1.142)	6.252 (0.957)	6.732 (1.117)	5.636 (0.0001)	0.1489

Abbreviations. IBD: Inflammatory Bowel Disease; DEXA: Dual Energy X-ray Absorptiometry; BIA: Bio-Impedensometric Analysis. SMI: Skeletal Muscle Index; ASMI: Appendicular Skeletal Mass Index.

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
