# Peer review of "Characterization of Sarcopenia in an IBD Population Attending an Italian Gastroenterology Tertiary Center"

_nutrients, 2019, doi:10.3390/nu11102281_

Round 1
Reviewer 1 Report
Pizzoferrato and colleagues report the percentage of sarcopenia in an IBD cohort. I have only some minor suggestions
Minor points.
Table 1. Specify the biologics used for the treatment
Figure 1 The term prevalence is not appropriate, because this is a rather small population. Please state instead percentage.
How many patents with sarcopenia were treated with seroids?
Author Response
Dear Reviewer 1,
thank you for your comments and suggestions. We proceeded to make the requested changes
Reviewer 2 Report
Authors presented very elegant and interesting study, which is highlighting the fact, that IBD are systemic diseases.
I have one small remark - please rewrite the second part in the conclusions in abstract. Should be more clear.
Author Response
Dear Reviewer 2,
thank you for your comments and suggestions. We have rewrited the second part in the conclusions in abstract.